# Cross-sectional prevalence study of MERS-CoV in local and imported dromedary camels in Saudi Arabia, 2016-2018

Ahmed M. Tolah[1,2], Saad B. AL Masaudi[2], Sherif A. El-Kafrawy[1,3], Ahmed A. Mirza[3], Steve M. Harakeh[1,3], Ahmed M. Hassan[1], Mohammed A. Alsaadi[1], Abdulrahman A. Alzahrani[4], Ghaleb A. Alsaaidi[4], Nabil M. S. Amor[5], Abdulaziz N. Alagaili[5], Anwar M. Hashem[6,7]*, Esam I. Azhar[1,3]*

1 King Fahd Medical Research Center, Special Infectious Agents Unit, King Abdulaziz University, Jeddah, Saudi Arabia, 2 Division of Microbiology, Department of Biological Science, Faculty of science, King Abdulaziz University, Jeddah, Saudi Arabia, 3 Department of Medical Laboratory Technology, Faculty of Applied Medical Sciences, King Abdulaziz University, Jeddah, Saudi Arabia, 4 Directorate of Agriculture, Ministry of Environment Water and Agriculture, Makkah Region, Saudi Arabia, 5 KSU Mammals Research Chair, Department of Zoology, College of Science, King Saud University, Riyadh, Saudi Arabia, 6 King Fahd Medical Research Center, Vaccines and Immunotherapy Unit, King Abdulaziz University, Jeddah, Saudi Arabia, 7 Department of Medical Microbiology and Parasitology, Faculty of Medicine, King Abdulaziz University, Jeddah, Saudi Arabia

* amhashem@kau.edu.sa (AH); eazhar@kau.edu.sa (EA)

**Data Availability Statement:** All relevant data are within the manuscript.

## Abstract

The Middle East Respiratory Syndrome-Coronavirus (MERS-CoV) is an endemic virus in dromedaries. Annually, Saudi Arabia imports thousands of camels from the Horn of Africa, yet the epidemiology of MERS-CoV in these animals is largely unknown. Here, MERS-CoV prevalence was compared in imported African camels and their local counterparts. A total of 1399 paired sera and nasal swabs were collected from camels between 2016 and 2018. Imported animals from Sudan (n = 829) and Djibouti (n = 328) were sampled on incoming ships at Jeddah Islamic seaport before unloading, and local camels were sampled from Jeddah (n = 242). Samples were screened for neutralizing antibodies (nAbs) and MERS-CoV viral RNA. The overall seroprevalence was 92.7% and RNA detection rate was 17.2%. Imported camels had higher seroprevalence compared to resident herds (93.8% vs 87.6%, p <0.01) in contrast to RNA detection (13.3% vs 35.5%, p <0.0001). Seroprevalence significantly increased with age (p<0.0001) and viral RNA detection rate was ~2-folds higher in camels <2-year-old compared to older animals. RNA detection was higher in males verses females (24.3% vs 12.6%, p<0.0001) but seroprevalence was similar. Concurrent positivity for viral RNA and nAbs was found in >87% of the RNA positive animals, increased with age and was sex-dependent. Importantly, reduced viral RNA load was positively correlated with nAb titers. Our data confirm the widespread of MERS-CoV in imported and domestic camels in Saudi Arabia and highlight the need for continuous active surveillance and better prevention measures. Further studies are also warranted to understand camels correlates of protection for proper vaccine development.

**Funding:** This work was supported by King Fahd Medical Research Center (KFMRC), King Abdulaziz University (KAU), Jeddah, Saudi Arabia, the Vice Deanship of Research Chairs at the King Saud University, Riyadh, Saudi Arabia, and King Abdulaziz City for Science and Technology (KACST), Riyadh, Saudi Arabia [grant number 09-1].The funders had no role in study design, data collection and analysis, decision to publish, or preparation of the manuscript.

**Competing interests:** The authors have declared that no competing interests exist.

## Introduction

Since its first isolation from a patient in Saudi Arabia in 2012, the Middle East respiratory syndrome-Coronavirus (MERS-CoV) has been causing human infections coupled with high fatality rates [1–5]. As of October 2019, 2468 confirmed human cases and 851 deaths (~34.5%) have been reported to the World Health Organization (WHO) from 27 countries [6]. Nonetheless, most of the cases and deaths are from Saudi Arabia, and all cases reported from Africa, Europe or North America are epidemiologically linked to the Arabian Peninsula [6]. While good control and prevention measures, when in place, limited MERS-CoV spread in healthcare and household settings, the majority of MERS cases are secondary due to human-to-human transmission in such locations [4,5,7,8].

Current epidemiological evidence suggest that dromedary camels are the only known virus reservoir and the main source of zoonotic transmission to humans [9,10] despite the reports suggesting a possible MERS-CoV circulation in other animal species such as sheep, goats, alpacas and llamas [11–14]. Indeed, several studies have linked human cases with exposure to infected camels where identical or nearly identical camel and human MERS-CoV isolates were detected [15–19]. Furthermore, seroprevalence rate in those who are in contact with camels is markedly higher than the general population [20–22], further indicating that camel exposure is a major risk factor [23].

Widespread active MERS infection was detected in dromedaries from 15 different countries in Asia (Saudi Arabia, United Arab Emirates, Qatar, Oman, Iraq, Jordan, Kuwait, Iran and Pakistan) and Africa (Egypt, Ethiopia, Kenya, Nigeria, Burkina Faso and Morocco) [9,10]. Interestingly, no viral RNA was found in camels from some Asian countries despite the seropositivity in their dromedary population [13,24]. Viral shedding in dromedaries usually lasts for 2 weeks but RNA positivity could persist for extended period of time and could be detected in stool and milk in addition to respiratory secretions [25–31].

Seropositivity in dromedaries ranged between ~30–100% in 20 Middle Eastern and African countries with lower rates being observed in the Canary Islands (4–9%) [9,10]. On the other hand, no MERS-CoV seropositive dromedaries were found in North America, Australia, Kazakhstan and Japan [9,10]. Similarly, there is no evidence of MERS-CoV circulation in bactrian camels in Mongolia and Kazakhstan [32–34]. Of note, presence of neutralizing antibodies (nAbs) in camels could only provide partial protection but not sterilizing immunity possibly due to exposure to different lineages of the virus and/or waning of nAbs [27–29,35–37]. Furthermore, it has been shown that increase in dromedary age is usually associated with increase in seroprevalence and decrease in positivity for MERS-CoV RNA [9,10].

Serological evidence suggests that MERS-CoV or related virus has been circulating in African dromedaries for >35 years [38], yet data on autochthonous MERS infections in humans in Africa are sparse mostly probably due to limited epidemiological surveillance and/or genetic divergence of African MERS-CoV lineages compared to those in the Arabian Peninsula [35,39]. Interestingly, African MERS-CoV lineages do not seem to be established or cause infections in camels or humans in Saudi Arabia despite the continued importation of African camels into the Arabian Peninsula [40].

Most imported camels in Saudi Arabia come from the Horn of Africa (Eritrea, Ethiopia, Djibouti, Somalia, Sudan, and Kenya). They are mostly received via Jeddah Islamic seaport (Western region of Saudi Arabia) and to a lesser extent at Jizan seaport (Southwestern region of Saudi Arabia) where they may or may not be quarantined for few days before their movement to local markets [41]. These imported camels are usually adult animals or at least >2-year-old, however, young and juvenile camels (<2-year-old) could also comprise a significant number of incoming shipments. However, the epidemiology of MERS-CoV in imported

dromedaries in Saudi Arabia is largely unknown. Therefore, the aim of this study was to investigate and compare the prevalence of MERS-CoV in imported camels from Sudan and Djibouti and local camels in Jeddah, Saudi Arabia by testing for nAbs and viral RNA to better evaluate the role of incoming animals in the introduction and epidemiology of the disease in Saudi Arabia and to help in implementing proper control measures.

## Materials and methods

### Samples

A total of 1399 paired blood samples and nasal swabs were collected from imported and local dromedary camels. Samples from imported camels (n = 1157) were collected between August 2016 and August 2018. These samples were randomly collected from ~15% of the animals in each animal compartment on incoming vessels from Sudan and Djibouti through Jeddah Islamic seaport before ships unloading. Samples from local camels (n = 242) were collected between March 2018 and August 2018 from 4 local farms north and south of Jeddah (30% of the sampled camels) and one local abattoir (70% of the sampled animals). All samples were transported in refrigerated containers for processing and analysis. Blood samples were collected from jugular vein in pre-labelled tubes which were centrifuged at 2,500 rpm for 5 min, and serum was separated, aliquoted and frozen at $-80^\circ$C until testing. Nasal swabs were collected using dacron swabs in viral transport medium (VTM) and stored at -80$^\circ$C until RNA extraction. Nasal swabs used in this study were from a larger sample pool used to study the genetic diversity of MERS-CoV in Saudi Arabia [40].

### RNA-extraction and reverse transcription-PCR-screening

Viral RNA was extracted from 200µl of the VTM sample using QiaAmp viral RNA extraction kit (Qiagen, Germany) according to the manufacturer's instructions. Testing for MERS-CoV was done using real-time reverse transcription-PCR (RT-PCR) assays targeting upE and ORF 1A as described previously [42]. Samples that were positive for both targets with cycle thresholds (Ct) values <40 were considered positive.

### Live virus microneutralization assay

Serum samples were tested for nAbs as described previously [43]. Briefly, heat inactivated sera were diluted at 1:20 and co-incubated with equal volume of media containing 100 $TCID_{50}$ of MERS-CoV for 1 h at 37$^\circ$C to test for nAbs. Serum-virus mixture was then transferred to confluent Vero E6 monolayer in a 96-well plate and incubated for three days at 37$^\circ$C in 5% $CO_2$ humidified incubator. Each sample was tested in quadruplicates and samples with inhibited cytopathic effect (CPE) in all wells were considered positive with nAb titer of $\geq$20. Around 20% of the seropositive serum samples from viral RNA shedding camels were randomly selected and serially diluted to determine nAb titers.

### Statistical analysis

Two-tail Chi$^2$ ($\chi^2$) test, Fisher exact test and 95% confidence intervals (CIs) were calculated using OpenEpi (Open Source Epidemiologic Statistics for Public Health). A p value of <0.05 was considered significant for all analyses.

### Ethics statement

The study was conducted after obtaining the needed permits and approvals from the Directorate of Agriculture, Ministry of Environment, Water and Agriculture, Jeddah, Saudi Arabia.

The study was approved by the Unit of Biomedical Ethics, King Abdulaziz University Hospital, (Approval number 16–121)

## Results

### Overall demography of sampled camels

Among the total samples (n = 1399), 82.7% (n = 1157) were collected from imported African dromedaries in which 71.7% (n = 829) and 28.3% (n = 328) were from Sudan and Djibouti, respectively (Fig 1). The remaining 17.3% of the samples (n = 242) were from local camels sampled in Jeddah. Sex and age were only known for 83.2% (n = 1164) of the sampled camels. Male camels represented 68.3% (n = 800) of the sampled animals compared to 31.3% (n = 364) females. Most camels were older than 2 years (61.3%, n = 713) as compared to younger animals between the age of 1–2 years (35.0%, n = 408) or those <1-year-old (3.7%, n = 43) as shown in Table 1. The details of camel population characteristics are described in a previously published article (REF).

### Seroprevalence of MERS-CoV in imported and local dromedaries in Saudi Arabia

Out of the 1399 tested sera, 92.7% (n = 1297) were seropositive with nAb titer ≥20. The seroprevalence was significantly higher (p <0.01) in imported camels (93.8%) compared to local animals (87.6%) (Fig 2). The highest seroprevalence was found in camels from Djibouti (97.9%, 95% CI 96.3–99.4) followed by those from Sudan (92.2%, 95% CI 90.3–94.0) compared to camels from Jeddah (87.6%, 95% CI 83.5–91.8) (Fig 2 and Table 1).

Seroprevalence significantly (p <0.0001) increased with age as lower rates were observed in juvenile camels (<1-year-old) compared those >1-year-old despite their origin (Fig 2). The highest seroprevalence rate was found in camels aged >2 years (96.5%, 95% CI 95.1–97.8) followed by those aged 1–2 years (86.5%, 95% CI 83.2–89.8) and <1 year (67.4%, 95% CI 53.4–81.5) (Table 1). However, there was no significant difference between local and imported camels from the same age group except for adult camels from Djibouti (97.7%) which were significantly higher (p <0.001) than adults from Sudan (91.5%) and Jeddah (89.8%) as shown in Fig 2.

Although higher seroprevalence was observed in females (94%, 95% CI 91.5–96.4) compared to males (91%, 95% CI 89.0–93.0), the overall differences did not reach significant levels for imported and local camels (Fig 2 and Table 1). However, the rate was significantly higher (p <0.001) in females compared to males in Sudanese camels only (Fig 2). There was also a significant difference (p <0.05) between local and imported male camels mostly because of the significantly higher rate (p <0.0001) in animals from Djibouti compared to those from Sudan and Jeddah (Fig 2). No statistical difference was found based on the year of collection (Table 1). Together, these data showed that most imported and local camels have been exposed to MERS-CoV in which seropositivity was significantly higher in imported and older animals. Details of seroprevalence from each of the three countries are shown in Table 2.

### MERS-CoV RNA prevalence in imported and local dromedaries in Saudi Arabia

Next, we determined the prevalence rate of MERS-CoV RNA. Viral RNA was detected in 240 nasal swabs (17.2%, 95% CI 15.2–19.1) of the total 1399 sampled animals (Fig 3 and Table 1), with significantly higher detection rate (p <0.0001) in local camels (Fig 3). Specifically, MERS-CoV RNA was detected in 35.5% (95% CI 29.5–41.6) of camels from Jeddah, in 14.1%

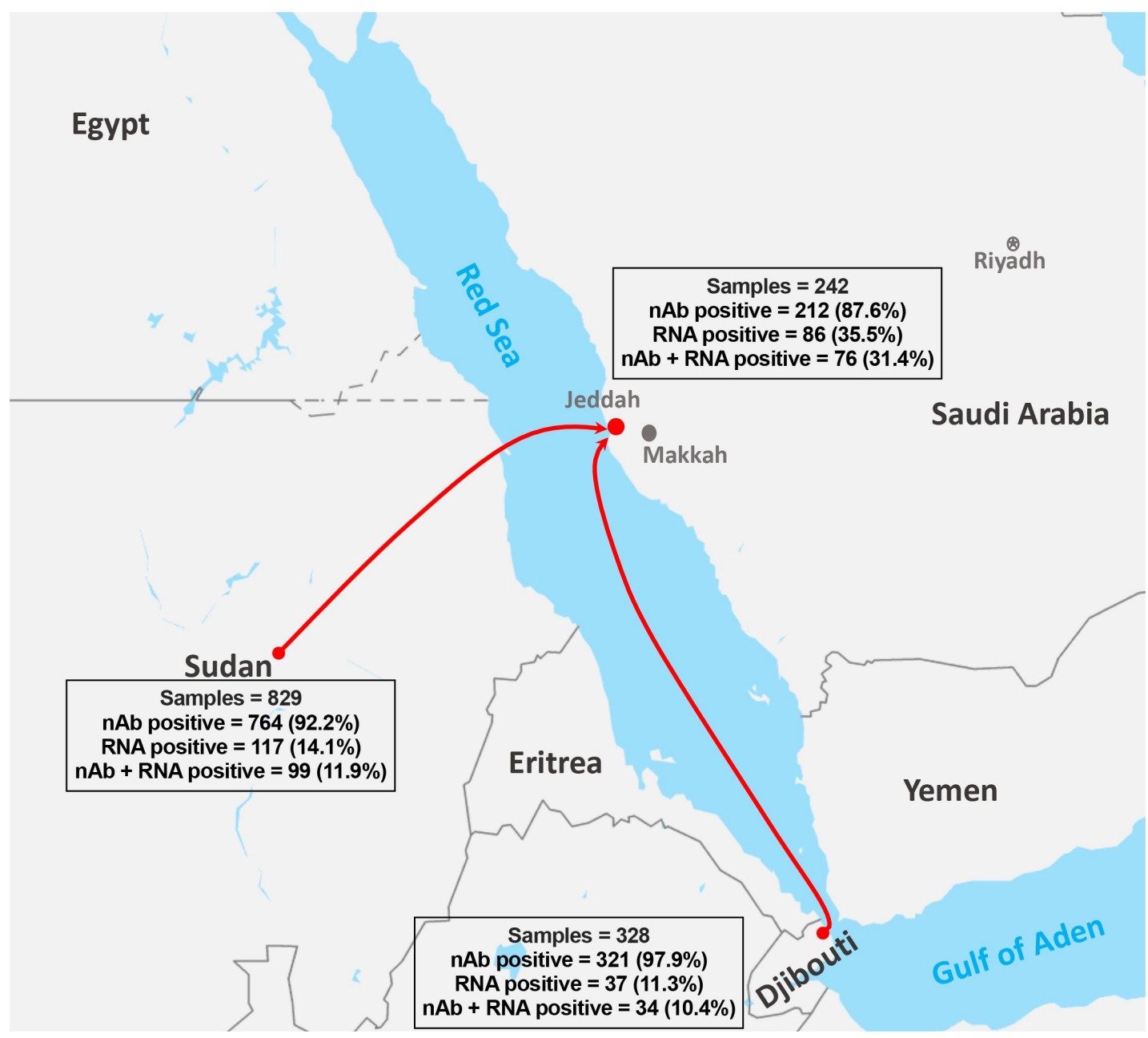

**Fig 1. Map of source of collected samples from imported and local dromedary camels.** Results from each source is shown as number and percentage of detected nAbs and viral RNA.

(95% CI 11.7–16.5) of camels from Sudan, and in 11.3% (95% CI 7.9–14.7) of camels from Djibouti (Fig 3 and Table 1). Genetic characteristics of the domestic and imported strains together with phylogenetic analysis are discussed in a previous publication [40].

Interestingly, RNA detection was markedly higher in camels aged 1–2 years (28.4%, 95% CI 24.1–32.8) compared to younger (9.3%, 95% CI 0.6–18.0) and older animals (16.8%, 95% CI 14.1–19.6) (Table 1). However, no significant difference between juvenile camels <1-year-old and adult animals >1-year-old was observed except for local dromedaries from Jeddah (p <0.01) as shown in Fig 3. While there was no significant difference in the detection rate of

**Table 1. Overall positivity for MERS-CoV nAbs and RNA in imported and local dromedaries in Saudi Arabia.**

| | Samples tested n (%) | MN Assay Positive n (%; 95% CI) | RT-PCR Positive n (%; 95% CI) | MN Assay & RT-PCR Positive n (%; 95% CI) |
|---|---|---|---|---|
| **Origin** | | | | |
| Djibouti | 328 (23.4) | 321 (97.9; 96.3–99.4) | 37 (11.3; 7.9–14.7) | 34 (10.4; 7.1–13.7) |
| Sudan | 829 (59.3) | 764 (92.2; 90.3–94.0) | 117 (14.1; 11.7–16.5) | 99 (11.9; 9.7–14.2) |
| Jeddah | 242 (17.3) | 212 (87.6; 83.5–91.8) | 86 (35.5; 29.5–41.6) | 76 (31.4; 25.6–37.3) |
| *p value* | | <0.0001 | <0.0001 | <0.0001 |
| **Age*** | | | | |
| < 1 year | 43 (3.1) | 29 (67.4; 53.4–81.5) | 4 (9.3; 0.6–18.0) | 4 (9.3; 0.6–18.0) |
| 1–2 years | 408 (29.2) | 353 (86.5; 83.2–89.8) | 116 (28.4; 24.1–32.8) | 91 (22.3; 18.3–26.3) |
| >2 year | 713 (51.0) | 688 (96.5; 95.1–97.8) | 120 (16.8; 14.1–19.6) | 114 (16.0; 13.3–18.7) |
| *p value* | | <0.0001 | <0.0001 | 0.0096 |
| **Sex*** | | | | |
| Male | 800 | 728 (91.0; 89.0–93.0) | 194 (24.3; 21.3–27.2) | 165 (20.6; 17.8–23.4) |
| Female | 364 | 342 (94.0; 91.5–96.4) | 46 (12.6; 9.2–16.1) | 44 (12.1; 8.7–15.4) |
| *p value* | | 0.0862 | <0.0001 | <0.0001 |
| **Year** | | | | |
| 2016 | 382 | 354 (92.7; 90.1–95.3) | 42 (11.0; 7.9–14.1) | 30 (7.9; 5.2–10.6) |
| 2017 | 637 | 595 (93.4; 91.5–95.3) | 73 (11.5; 9.0–13.9) | 68 (10.7; 8.3–13.1) |
| 2018 | 380 | 348 (91.6; 88.8–94.4) | 125 (32.9; 28.2–37.6) | 111 (29.2; 24.6–33.8) |
| *p value* | | 0.5551 | <0.0001 | 0.0004 |
| **Total** | **1399** | **1297 (92.7; 91.4–94.1)** | **240 (17.2; 15.2–19.1)** | **209 (14.9; 13.1–16.8)** |

*Sex and age were only available for 1164 samples (83.2%) out of the 1399 tested samples.

MERS-CoV in local and imported juveniles, rates of viral RNA detection from local adults (>1-year-old) was significantly higher (p <0.0001) when compared to imported adult camels from Sudan and Djibouti (Fig 3).

Male camels always had significantly higher detection rate (p <0.0001) of MERS-CoV compared to females despite their origin except for Djibouti as no females were sampled (Fig 3 and Table 1). Consistent with the overall higher rate in local camels, viral RNA was detected in male and female resident animals at significantly higher rates (p <0.0001 and p <0.05, respectively) when compared to imported camels (Fig 3). Furthermore, significantly higher number (p <0.0001) of male camels from Sudan (26.7%) tested positive for MERS-CoV compared to males from Djibouti (12.1%) (Fig 3). While we observed significant difference (p <0.0001) in MERS-CoV RNA detection based on the year of collection (Table 1), these results could be biased because of the fact that only camels from Sudan were sampled during the three years and those from Djibouti and Jeddah were sampled only during 2018 (Table 2).

## Concurrent positivity for MERS-CoV RNA and nAbs

We found that 209 of the total camels (14.9%; 95% CI 13.1–16.8) were positive for both MERS-CoV RNA and nAbs which represents >87% of the RNA positive animals (Table 1), suggesting that these camels were seroconverted, re-infected or have maternal nAbs. Indeed, 100% of the juvenile camels <1-year-old that were MERS-CoV RNA positive were also sero-positive (Table 1), most probably due to acquiring maternal nAbs. Apart from that, concurrent positivity for viral RNA and nAbs increased with age as 78.5% and 95.0% of the RNA positive camels aged 1–2 years and >2 years were also seropositive, respectively (Table 1). Sex was also a factor in dual positivity among the RNA positive camels with 95.7% (44 out of 46) and 85.1% (165 out of 194) prevalence in females and males, respectively.

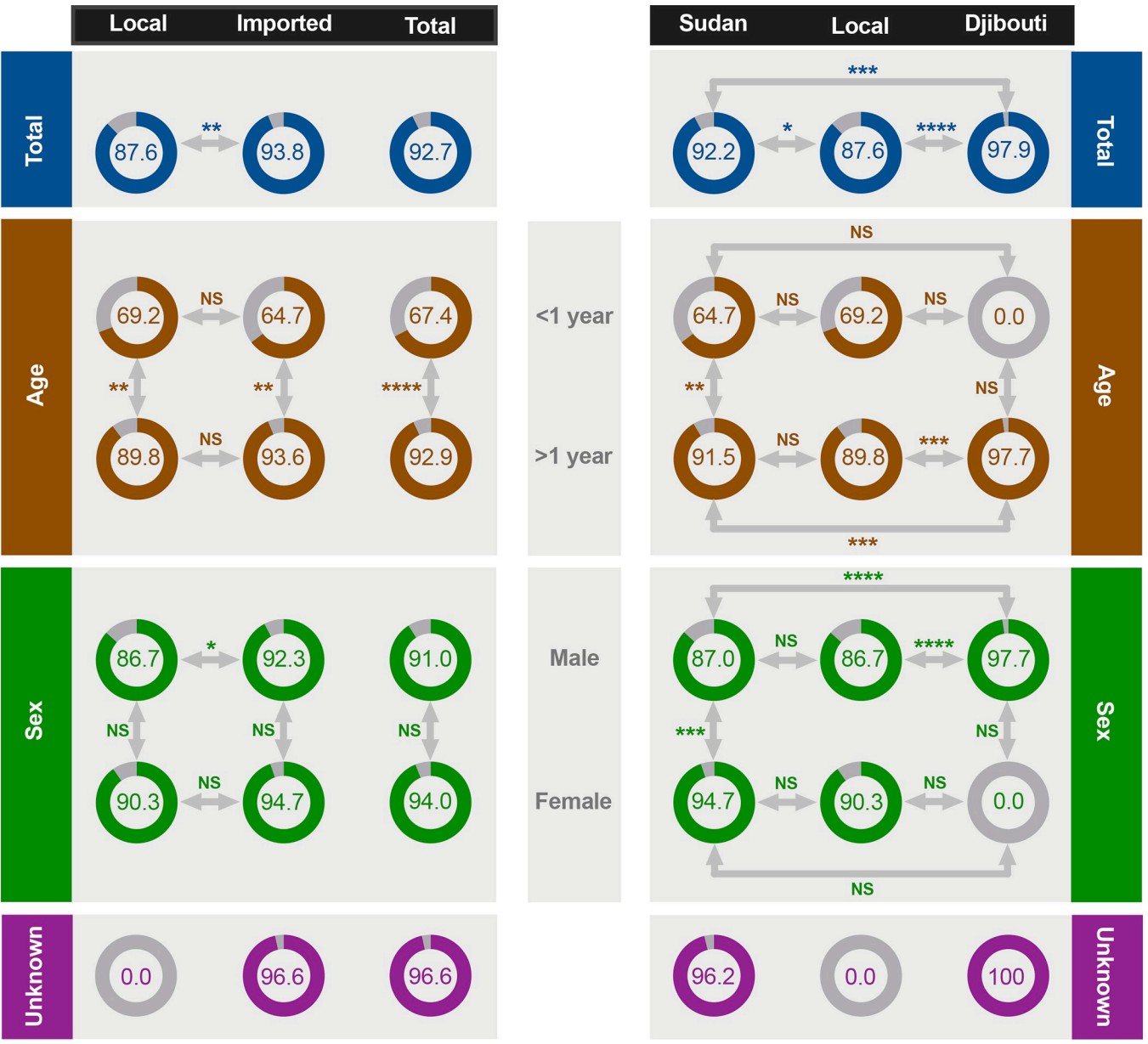

**Fig 2. Seroprevalence rate in imported and local dromedary camels based on age and sex.** Percentage of seroprevalence is shown for each category.
****p<0.0001, ***p <0.001, **p <0.01, *p <0.05 and NS denotes not statistically significant (Fisher Exact Test).

No significant differences in viral load (as evaluated by comparing their Ct values in the real time RT-PCR assay) were found based on sex or age except for adult camels >2-year-old compared to those aged between 1–2 years (p <0.001) as shown in Fig 4A and 4B. Of note, significantly higher viral load (p <0.0001) was observed in seronegative camels (mean Ct value ~25.8) compared to seropositive animals (mean Ct value ~33.0) as shown in Fig 4C. Furthermore, we determined the nAb titers in 40 samples from those positive for both viral RNA and nAbs and found that the overall geometric mean titer of nAbs was 8.5±1.3 log2 (range 4.3–13.3 log2). These titers showed strong positive correlation with Ct values (Pearson correlation coefficient 0.5602, p = 0.0003).

**Table 2. Positivity for MERS-CoV nAbs and RNA in camels based on age, sex and year of collection for each individual country.**

| | | Samples tested n (%) | MN Assay Positive n (%) | RT-PCR Positive n (%) | MN Assay & RT-PCR Positive n (%) |
|---|---|---|---|---|---|
| **Sudan** | **Age**[*] | | | | |
| | < 1 year | 17 (2.1) | 11 (64.7) | 1 (5.9) | 1 (5.9) |
| | 1–2 years | 304 (36.7) | 263 (86.5) | 83 (27.3) | 66 (21.7) |
| | >2 year | 296 (35.7) | 286 (96.6) | 33 (11.1) | 32 (10.8) |
| | Unknown | 212 (25.6) | 204 (96.2) | 0 (0.0) | 0 (0.0) |
| | **Sex**[*] | | | | |
| | Male | 315 (38.0) | 274 (87.0) | 84 (26.7) | 67 (21.3) |
| | Female | 302 (36.4) | 286 (94.7) | 33 (10.9) | 32 (10.6) |
| | Unknown | 212 (25.6) | 204 (96.2) | 0 (0.0) | 0 (0.0) |
| | **Year** | | | | |
| | 2016 | 359 (43.3) | 331 (92.2) | 42 (11.7) | 30 (8.4) |
| | 2017 | 427 (51.5) | 392 (91.8) | 73 (17.1) | 68 (15.9) |
| | 2018 | 43 (5.2) | 41 (95.3) | 2 (4.7) | 1 (2.3) |
| | **Total** | **829** | **764 (92.2)** | **117 (14.1)** | **99 (11.9)** |
| **Djibouti** | **Age**[*] | | | | |
| | < 1 year | 0 (0.0) | 0 (0.0) | 0 (0.0) | 0 (0.0) |
| | 1–2 years | 25 (7.6) | 24 (96.0) | 0 (0.0) | 0 (0.0) |
| | >2 year | 280 (85.4) | 274 (97.9) | 37 (13.2) | 34 (12.1) |
| | Unknown | 23 (7.0) | 23 (100.0) | 0 (0.0) | 0 (0.0) |
| | **Sex**[*] | | | | |
| | Male | 305 (93.0) | 298 (97.7) | 37 (12.1) | 34 (11.1) |
| | Female | 0 (0.0) | 0 (0.0) | 0 (0.0) | 0 (0.0) |
| | Unknown | 23 (7.0) | 23 (100.0) | 0 (0.0) | 0 (0.0) |
| | **Year** | | | | |
| | 2016 | 23 (7.0) | 23 (100.0) | 0 (0.0) | 0 (0.0) |
| | 2017 | 210 (64.0) | 203 (96.7) | 0 (0.0) | 0 (0.0) |
| | 2018 | 95 (29.0) | 95 (100.0) | 37 (38.9) | 34 (35.8) |
| | **Total** | **328** | **321 (97.9)** | **37 (11.3)** | **34 (10.4)** |
| **Jeddah** | **Age**[*] | | | | |
| | < 1 year | 26 (10.7) | 18 (69.2) | 3 (11.5) | 3 (11.5) |
| | 1–2 years | 79 (32.6) | 66 (83.5) | 33 (41.8) | 25 (31.6) |
| | >2 year | 137 (56.6) | 128 (93.4) | 50 (36.5) | 48 (35.0) |
| | Unknown | 0 (0.0) | 0 (0.0) | 0 (0.0) | 0 (0.0) |
| | **Sex**[*] | | | | |
| | Male | 180 (74.4) | 156 (86.7) | 73 (40.6) | 64 (35.6) |
| | Female | 62 (25.6) | 56 (90.3) | 13 (21.0) | 12 (19.4) |
| | Unknown | 0 (0.0) | 0 (0.0) | 0 (0.0) | 0 (0.0) |
| | **Year** | | | | |
| | 2016 | 0 (0.0) | 0 (0.0) | 0 (0.0) | 0 (0.0) |
| | 2017 | 0 (0.0) | 0 (0.0) | 0 (0.0) | 0 (0.0) |
| | 2018 | 242 (100.0) | 212 (87.6) | 86 (35.5) | 76 (31.4) |
| | **Total** | **242** | **212 (87.6)** | **86 (35.5)** | **76 (31.4)** |

## Discussion

Our serological results indicate a widespread distribution of MERS-CoV in dromedaries from all tested countries where most imported and local camels (92.7%) showed an evidence of previous exposure to MERS-CoV. Seropositivity was significantly higher in imported compared

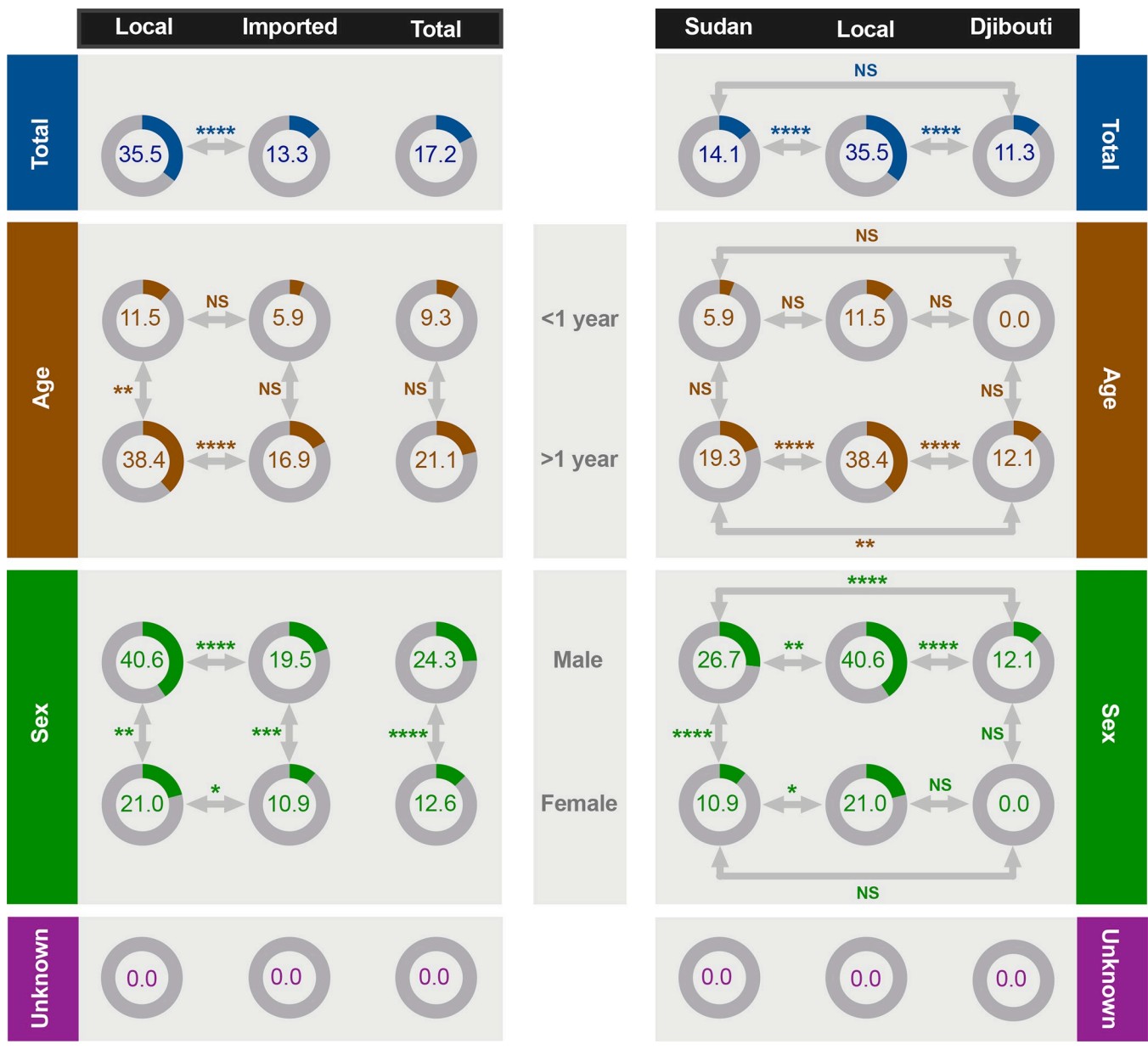

**Fig 3. RNA detection rate in imported and local dromedary camels based on age and sex.** Percentage of seroprevalence is shown for each category. ****p<0.0001, ***p <0.001, **p <0.01, *p <0.05 and NS denotes not statistically significant (Fisher Exact Test).

to local dromedaries consistent with most previous studies [9,10,12,24,29]. The highest seroprevalence was found in camels from Djibouti (97.9%), which is reported for the first time, followed by those from Sudan (92.2%) compared to local camels from Jeddah (87.6%).

On the contrary, imported camels had significantly lower MERS-CoV RNA (11.3% and 14.1% in camels from Djibouti and Sudan, respectively) compared to locally sampled animals (35.5%). This is in accordance with a previous report from Saudi Arabia [44] but not those from Egypt [12,29,45]. However, all these previous studies have sampled imported camels at abattoirs, markets and quarantines compared to camels in our study which were sampled on the incoming individual ships before unloading and exposure to other camels. Therefore,

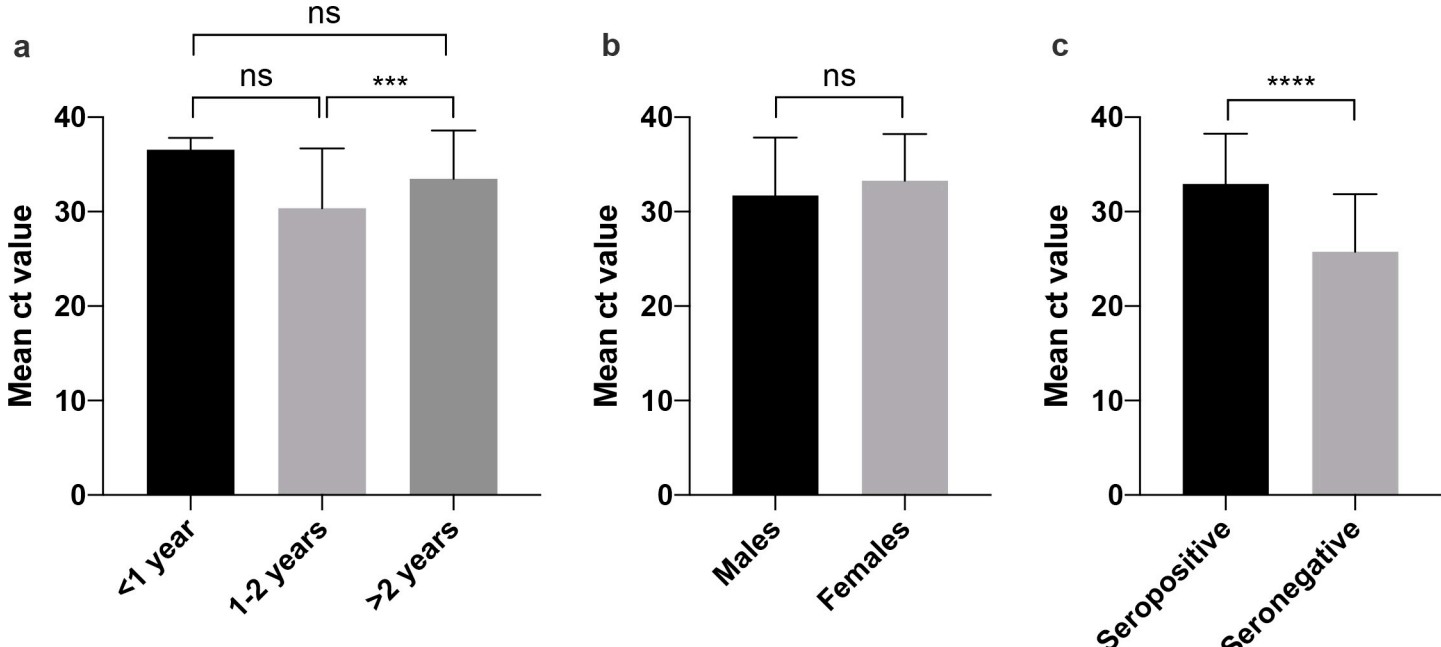

**Fig 4. Comparison of MERS-CoV RNA load based on age, sex and presence of nAbs.** One-way ANOVA with Bonferroni post-test was used for (a) and student's t test was used in (b and c). ****p<0.0001, ***p <0.001 and NS denotes not statistically significant.

some uncertainties of these previous findings remain regarding the extent of exposure of imported camels to their local counterpart at such sites and the possible amplification of the infection at the end of the value-chain and before the actual sampling. It is noteworthy that the rate of positivity for MERS-CoV RNA in local camels is higher than those reported by most previous studies from Saudi Arabia which ranged from 12.1%-29.0% [18,44,46–48] except for one study which showed a higher rate of 56.0% [19]. Furthermore, while temporal viral shedding has been suggested between December to May [19,29], viral RNA was detected throughout the sampling period in the current study except for the months of July and August in imported camels (data not shown) as previously reported [44].

Previous studies have shown contradicting data on the effect of sex on seropositivity and presence of viral RNA [12,18,29,49–51]. Here, we observed higher seroprevalence rates in females compared to males, but these differences were not significant for both imported and local camels except for camels from Sudan. However, there was more significantly seropositive imported males compared to local males. Detection rate of MERS-CoV was significantly higher in males vs females despite their origin consistent with most previous reports [18,29,51] except for one study from Egypt [12]. Nonetheless, local male and female camels were significantly more positive for viral RNA compared to imported animals.

Significant incremental increase in seroprevalence was observed in older camels in this study in which the overall rates increased from 67.4% in camels aged <1 year to 86.5% and 96.5% in those 1-2-year-old and >2-year-old, respectively. Similar age dependent pattern of seroprevalence was reported for MERS-CoV in camels in most other studies [9,10]. No significant difference was found in seropositivity between local and imported camels from the same age group except for adults from Djibouti which were significantly higher than their counterparts from Sudan and Saudi Arabia. Of note that camels from Djibouti were all >1-year-old. While no differences were observed between those aged >2 years from Djibouti (97.9%) compared to animals from Sudan or Saudi Arabia (96.6% and 93.4%, respectively), camels in the

age range of 1–2 years were significantly more seropositive in Djibouti (96.0%) when compared to those from Sudan (86.5%) and Saudi Arabia (83.5%) (Table 2).

Several studies have shown an inverse association between viral RNA positivity and age in dromedaries in countries in Africa and the Middle East [18,19,36,46,51], although no significant differences were found in some studies from Egypt [12,29]. In contrast, we found that the overall rate of viral RNA positivity in camels >1-year-old (21.1%) is significantly higher than that in dromedaries <1-year-old (9.3%). Interestingly, RNA detection was significantly higher in camels aged 1–2 years with an overall rate of 28.4% compared to 9.3% and 16.8% in younger and older animals, respectively (Table 1). This pattern was clearly observed when age stratified data from each country were examined in which 41.8% and 27.3% of the 1-2-year-old camels were positive compared to 36.5% and 11.1% of the >2-year-old camels and 11.5% and 5.9% of the <1-year-old animals from Saudi Arabia and Sudan, respectively (Table 2). Notably, none of the 1-2-year-old camels from Djibouti had any evidence of active MERS-CoV infection compared to those >2-year-old. Together, these data show that there are almost 2-fold higher infection rate in camels <2 years compared to older camels suggesting that MERS-CoV infection mostly occurs in young camels and could represent a major source of human infections. Also, the presence of naïve young camels in such environment or at any stage in the value-chain could contribute to virus maintenance and amplification due to co-mixing or co-housing.

More than 87% of viral RNA shedding dromedaries were also seropositive for nAbs in which such camels were detected from all three countries. Generally, concurrent rate of positivity for viral RNA and nAbs was age-dependent (78.5% and 95.0% in camels aged 1–2 years and >2 years, respectively) except for camels <1-year-old which showed 100% positivity form both RNA and nAbs. While the presence of nAbs in animals >1-year-old could be due to sero-conversion during early stage of infection, intermittent viral RNA shedding or re-infection possibly with different MERS-CoV lineages, the 100% positivity for MERS-CoV RNA and nAbs in juvenile camels suggest that these nAbs could be maternal rather than acquired due to infection. However, the possibility of infection of juvenile camels cannot be excluded completely. These data are in accordance with several previous studies which found nAbs and viral RNA shedding in both calves and older dromedaries [12,36,37]. Sex was also a determinant factor in co-positivity. We observed significantly more positive males vs females for both RNA and nAbs consistent with the overall higher rate of RNA shedding in males and almost similar seropositivity. However, females positive for MERS-CoV RNA were more prone to be seropositive (95.7%) compared to males (85.1%).

While the role of nAbs in preventing infection is still not fully clear especially with the observed decline in their titers over time [29,36,37], some existing evidence suggest that these Abs could minimize viral shedding and that sterile immunity could only be achieved at very high levels of nAbs [30,36,37]. Indeed, we found that seronegative camels tend to have significantly higher viral load compared to seropositive animals. Furthermore, we found a strong correlation between nAb titers and viral load in the cohort of camels that were positive for both viral RNA and nAbs. These data clearly suggest that while presence of nAbs could provide partial protection in camels and reduce viral load, our data point towards the fact that even high nAb titers cannot confer sterilizing immunity and prevent MERS-CoV re-infection in dromedaries.

In summary, our data show large geographical distribution and high endemicity of MERS-CoV in camels as most imported and local camels had serological evidence of exposure to the virus. Significantly higher seropositivity was found in imported and adult animals, and increased viral RNA excretion was detected in local, young (<2-year-old) and male camels. Nonetheless, MERS-CoV circulation was not uncommon among imported or adult camels.

Furthermore, our findings clearly show that the presence of high titers of nAbs is not necessarily sufficient to prevent MERS-CoV infections in dromedaries. Although the sampling time frames of imported and domestic camels are different, as imported camels were sampled between 2016–2018 while domestic camels were only sampled in 2018, the results are still indicative of the prevalence of MERS-CoV in both groups and require further investigation and continuous screening programs to evaluate the seasonality of the infection.

These findings have important implications with respect to prevention, epidemiology of the disease and vaccination. First, most imported camels are used for meat consumption and they are destined to fattening farms and/or holding yards and ultimately to abattoirs and markets where increased viral shedding has been reported [44,48,51]. The presence of naïve as well as viral shedding imported camels during co-housing or co-mixing in such locations with resident animals would certainly increase the chance of virus maintenance and amplification. This could also contribute to co-infection and may lead to recombination events between different viral lineages. Therefore, better prevention and countermeasures should be applied not only to minimize virus circulation in camels but also to reduce any possible zoonotic transmission to high risk groups such as abattoir workers, shepherds, veterinary doctors, and camel handlers. Second, continuous systematic and longitudinal sampling and testing should be an integral part of any existing surveillance program to better understand the epidemiology of the disease, the dynamics of the infection and the various risk factors. Third, more studies are obviously needed to understand the correlates of protection in camels for better application and implementation of camel vaccination strategies as a one-health approach.

## Acknowledgments

The authors are extremely grateful to Dr. Hamad Albatshan, Deputy Minister for Animal Resources, Ministry of Environment water and Agriculture, Saudi Arabia for his unlimited and enthusiastic support to undertake this study. We would like also to thank King Fahd Medical Research Center (KFMRC), King Abdulaziz University (KAU) for the support. Also, we wish to thank the King Abdulaziz City for Science and Technology (KACST) for their generous funding through the MERS-CoV research grant program (number 09–1), which is a part of the Targeted Research Program (TRP).

## Author Contributions

**Conceptualization:** Saad B. AL Masaudi, Sherif A. El-Kafrawy, Esam I. Azhar.

**Data curation:** Sherif A. El-Kafrawy, Ahmed A. Mirza, Ahmed M. Hassan, Anwar M. Hashem, Esam I. Azhar.

**Formal analysis:** Anwar M. Hashem, Esam I. Azhar.

**Funding acquisition:** Esam I. Azhar.

**Investigation:** Esam I. Azhar.

**Methodology:** Ahmed M. Tolah, Sherif A. El-Kafrawy, Ahmed M. Hassan, Mohammed A. Alsaadi, Abdulrahman A. Alzahrani, Ghaleb A. Alsaaidi, Nabil M. S. Amor, Abdulaziz N. Alagaili, Esam I. Azhar.

**Project administration:** Esam I. Azhar.

**Resources:** Esam I. Azhar.

**Supervision:** Esam I. Azhar.

**Validation:** Esam I. Azhar.

**Visualization:** Esam I. Azhar.

**Writing – original draft:** Sherif A. El-Kafrawy.

**Writing – review & editing:** Saad B. AL Masaudi, Sherif A. El-Kafrawy, Ahmed A. Mirza, Steve M. Harakeh, Abdulaziz N. Alagaili, Anwar M. Hashem, Esam I. Azhar.

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
