## [Decision Letter · Decision Letter 0]

17 Feb 2020

PONE-D-20-01538

Cross-sectional prevalence study of MERS-CoV in local and imported dromedary camels in Saudi Arabia, 2016-2018

PLOS ONE

Dear Prof Azhar,

Thank you for submitting your manuscript to PLOS ONE. After careful consideration, we feel that it has merit but does not fully meet PLOS ONE’s publication criteria as it currently stands. Therefore, we invite you to submit a revised version of the manuscript that addresses the points raised during the review process.

We would appreciate receiving your revised manuscript by Apr 02 2020 11:59PM. To enhance the reproducibility of your results, we recommend that if applicable you deposit your laboratory protocols in protocols.io, where a protocol can be assigned its own identifier (DOI) such that it can be cited independently in the future. For instructions see: http://journals.plos.org/plosone/s/submission-guidelines#loc-laboratory-protocols

We look forward to receiving your revised manuscript.

Kind regards,

Stefan Pöhlmann, Ph.D.

Academic Editor

PLOS ONE

Journal Requirements:

3. We note that Figure #1 in your submission contain map images which may be copyrighted. All PLOS content is published under the Creative Commons Attribution License (CC BY 4.0), which means that the manuscript, images, and Supporting Information files will be freely available online, and any third party is permitted to access, download, copy, distribute, and use these materials in any way, even commercially, with proper attribution. For these reasons, we cannot publish previously copyrighted maps or satellite images created using proprietary data, such as Google software (Google Maps, Street View, and Earth). For more information, see our copyright guidelines: http://journals.plos.org/plosone/s/licenses-and-copyright.

a.    You may seek permission from the original copyright holder of Figure #1 to publish the content specifically under the CC BY 4.0 license. 

Reviewers' comments:

Reviewer's Responses to Questions

**Comments to the Author**

1. Is the manuscript technically sound, and do the data support the conclusions?

Reviewer #1: Yes

Reviewer #2: Yes

2. Has the statistical analysis been performed appropriately and rigorously? 

Reviewer #1: Yes

Reviewer #2: Yes

3. Have the authors made all data underlying the findings in their manuscript fully available?

Reviewer #1: Yes

Reviewer #2: Yes

4. Is the manuscript presented in an intelligible fashion and written in standard English?

Reviewer #1: Yes

Reviewer #2: Yes

5. Review Comments to the Author

Reviewer #1: In the present manuscript Tolah and coworkers present their findings from a comparative screening study for MERS-CoV neutralizing antibodies and viral nucleic acid in imported dromedaries from Sudan and Djibouti and in local dromedary camels in Jeddah/Saudi Arabia.

While local cohorts of dromedaries on the Arabian Peninsula had been extensively surveyed before, less information is available for dromedaries imported from Africa. Few studies have been conducted on assessing the prevalence of MERS-CoV-specific antibodies and MERS-CoV RNA in local dromedary camel herds in different African countries that export animals to the Arabian Peninsula. Moreover, data on MERS-CoV prevalence in imported dromedaries have so far only been obtained for animals held at quarantines, abattoirs and markets, and thus leave the possibility for (i) infection of imported animals by local MERS-CoV variants and (ii) for exchange of MERS-CoV variants between dromedaries imported from different sources. As a result, it is difficult to tell whether imported dromedaries that are tested MERS-CoV-positive were already infected/antibody-positive before being imported to the Arabian Peninsula or if infection occurred after import.

Tolah and coworkers analyzed samples taken during the transport of dromedaries from the Horn of Africa (countries: Sudan and Djibouti) to Saudi Arabia (Jeddah) in order to assess the prevalence of MERS-CoV in dromedaries before they touch ground in Saudi Arabia and might get in contact with other imported animals (from other African countries) or local animals. The authors report that imported dromedaries had significantly higher seroprevalence than local animals. In contrast, prevalence of MERS-CoV RNA was significantly higher in local dromedaries. Interestingly, the authors found gender-specific differences in the MERS-CoV RNA levels, an observation that has been also made by other groups. Further, a link between age and seropositivity was observed.

This work fills a gap of our knowledge on MERS-CoV prevalence in dromedary camels that are imported from Africa to the Arabian Peninsula. The manuscript is well-written and the data are clearly presented. Overall, I can recommend this manuscript for publication without further modifications.

Reviewer #2: The mansucript by Tolah et al compares MERS-CoV seroprevalence and viral RNA detection rates between imported and local dromedaries in Saudi Arabia between 2016 and 2018. The authors used previously established PCR and neutralization tests. The main findings are that imported camels had higher seroprevalence compared to resident animal in contrast to RNA detection. Seroprevalence significantly increased with age and viral RNA detection rate was 2-folds higher in camels <2 year-old compared to older animals. Interestingly, concurrent positivity for viral RNA and nAbs was found in >87% of the RNA positive animals which has implications for vaccination strategies.

The manuscript is overall well written and the data support the conclusions. The fact, that many dromedaries with neutralizing antibodies were also RNA positive confirms previous observations. Although the novelty might not be striking, the direct comparison of seropositivity and MERS-CoV RNA among imported vs resident dromedaries has not been studied in detail.

I have a few remarks:

1. The authors should emphasize more that a previous study was already based on the same sampling (El-Kafrawy, Lancet Planetary Health, 2019). Otherwise, readers might wonder why the authors did not include phylogenies etc.

2. The authors state in line 146 that they left the serum-virus mix on the cells for 3 days. This seems uncommon, please check.

3. In Figure 4 the authors compare ct values. How did the authors guarantee that all PCRs had the same efficiency? Did they use a control RNA, or an in vitro transcript as standard?

4. African dromedaries were sampled 2016 until 2018. The resident dromedary samples were exclusively collected in 2018. A direct comparison of e.g. RNA detection rates between African and Arabian dromedaries is therefore only possible for the year 2018. This limitation should be addressed.

6. PLOS authors have the option to publish the peer review history of their article (what does this mean?). If published, this will include your full peer review and any attached files.

Reviewer #1: No

Reviewer #2: No

---

## [Author Response · Author response to Decision Letter 0]

4 Apr 2020

Journal Requirements:

The manuscript was formatted to meet PLOS ONE style requirements.

The phrase was removed.

3. We note that Figure #1 in your submission contain map images which may be copyrighted. All PLOS content is published under the Creative Commons Attribution License (CC BY 4.0), which means that the manuscript, images, and Supporting Information files will be freely available online, and any third party is permitted to access, download, copy, distribute, and use these materials in any way, even commercially, with proper attribution. For these reasons, we cannot publish previously copyrighted maps or satellite images created using proprietary data, such as Google software (Google Maps, Street View, and Earth). For more information, see our copyright guidelines: http://journals.plos.org/plosone/s/licenses-and-copyright. We require you to either (a) present written permission from the copyright holder to publish these figures specifically under the CC BY 4.0 license, or (b) remove the figures from your submission:

a. You may seek permission from the original copyright holder of Figure #1 to publish the content specifically under the CC BY 4.0 license. We recommend that you contact the original copyright holder with the Content Permission Form (http://journals.plos.org/plosone/s/file?id=7c09/content-permission-form.pdf) and the following text: “I request permission for the open-access journal PLOS ONE to publish XXX under the Creative Commons Attribution License (CCAL) CC BY 4.0 (http://creativecommons.org/licenses/by/4.0/). Please be aware that this license allows unrestricted use and distribution, even commercially, by third parties. Please reply and provide explicit written permission to publish XXX under a CC BY license and complete the attached form.” Please upload the completed Content Permission Form or other proof of granted permissions as an "Other" file with your submission. In the figure caption of the copyrighted figure, please include the following text: “Reprinted from [ref] under a CC BY license, with permission from [name of publisher], original copyright [original copyright year].”

b. If you are unable to obtain permission from the original copyright holder to publish these figures under the CC BY 4.0 license or if the copyright holder’s requirements are incompatible with the CC BY 4.0 license, please either i) remove the figure or ii) supply a replacement figure that complies with the CC BY 4.0 license. Please check copyright information on all replacement figures and update the figure caption with source information. If applicable, please specify in the figure caption text when a figure is similar but not identical to the original image and is therefore for illustrative purposes only. The following resources for replacing copyrighted map figures may be helpful: USGS National Map Viewer (public domain): http://viewer.nationalmap.gov/viewer/ The Gateway to Astronaut Photography of Earth (public domain): http://eol.jsc.nasa.gov/sseop/clickmap/ Maps at the CIA (public domain): https://www.cia.gov/library/publications/the-world-factbook/index.htmland
https://www.cia.gov/library/publications/cia-maps-publications/index.html NASA Earth Observatory (public domain): http://earthobservatory.nasa.gov/ Landsat: http://landsat.visibleearth.nasa.gov/ USGS EROS (Earth Resources Observatory and Science (EROS) Center) (public domain): http://eros.usgs.gov/# Natural Earth (public domain): http://www.naturalearthdata.com/

The map was generated using a freely available victor mapping web site that does not require copy rights or licensing and we wrote the data on the generated map. 

 

Reviewers' comments:

Reviewer's Responses to Questions

Comments to the Author

1. Is the manuscript technically sound, and do the data support the conclusions?

Reviewer #1: Yes

Reviewer #2: Yes

2. Has the statistical analysis been performed appropriately and rigorously? 

Reviewer #1: Yes

Reviewer #2: Yes

3. Have the authors made all data underlying the findings in their manuscript fully available?

Reviewer #1: Yes

Reviewer #2: Yes

4. Is the manuscript presented in an intelligible fashion and written in standard English?

Reviewer #1: Yes

Reviewer #2: Yes

 

5. Review Comments to the Author

Reviewer #1: 

In the present manuscript Tolah and coworkers present their findings from a comparative screening study for MERS-CoV neutralizing antibodies and viral nucleic acid in imported dromedaries from Sudan and Djibouti and in local dromedary camels in Jeddah/Saudi Arabia.

While local cohorts of dromedaries on the Arabian Peninsula had been extensively surveyed before, less information is available for dromedaries imported from Africa. Few studies have been conducted on assessing the prevalence of MERS-CoV-specific antibodies and MERS-CoV RNA in local dromedary camel herds in different African countries that export animals to the Arabian Peninsula. Moreover, data on MERS-CoV prevalence in imported dromedaries have so far only been obtained for animals held at quarantines, abattoirs and markets, and thus leave the possibility for (i) infection of imported animals by local MERS-CoV variants and (ii) for exchange of MERS-CoV variants between dromedaries imported from different sources. As a result, it is difficult to tell whether imported dromedaries that are tested MERS-CoV-positive were already infected/antibody-positive before being imported to the Arabian Peninsula or if infection occurred after import.

Tolah and coworkers analyzed samples taken during the transport of dromedaries from the Horn of Africa (countries: Sudan and Djibouti) to Saudi Arabia (Jeddah) in order to assess the prevalence of MERS-CoV in dromedaries before they touch ground in Saudi Arabia and might get in contact with other imported animals (from other African countries) or local animals. The authors report that imported dromedaries had significantly higher seroprevalence than local animals. In contrast, prevalence of MERS-CoV RNA was significantly higher in local dromedaries. Interestingly, the authors found gender-specific differences in the MERS-CoV RNA levels, an observation that has been also made by other groups. Further, a link between age and seropositivity was observed.

This work fills a gap of our knowledge on MERS-CoV prevalence in dromedary camels that are imported from Africa to the Arabian Peninsula. The manuscript is well-written and the data are clearly presented. Overall, I can recommend this manuscript for publication without further modifications.

We thank the reviewer for the positive feedback.

 

Reviewer #2: 

The manuscript by Tolah et al compares MERS-CoV seroprevalence and viral RNA detection rates between imported and local dromedaries in Saudi Arabia between 2016 and 2018. The authors used previously established PCR and neutralization tests. The main findings are that imported camels had higher seroprevalence compared to resident animal in contrast to RNA detection. Seroprevalence significantly increased with age and viral RNA detection rate was 2-folds higher in camels <2 year-old compared to older animals. Interestingly, concurrent positivity for viral RNA and nAbs was found in >87% of the RNA positive animals which has implications for vaccination strategies.

The manuscript is overall well written and the data support the conclusions. The fact, that many dromedaries with neutralizing antibodies were also RNA positive confirms previous observations. Although the novelty might not be striking, the direct comparison of seropositivity and MERS-CoV RNA among imported vs resident dromedaries has not been studied in detail.

I have a few remarks:

1. The authors should emphasize more that a previous study was already based on the same sampling (El-Kafrawy, Lancet Planetary Health, 2019). Otherwise, readers might wonder why the authors did not include phylogenies etc.

The Following statement was added in the results section page 9, line 202-203 “Genetic characteristics of the domestic and imported strains together with phylogenetic analysis are discussed in a previous publication [40].”

2. The authors state in line 146 that they left the serum-virus mix on the cells for 3 days. This seems uncommon, please check.

We followed the protocol by our group in “Evaluation of MERS-CoV Neutralizing Antibodies in Sera Using Live Virus Microneutralization Assay” by Abdullah Algaissi and Anwar M. Hashem Book series: Methods In Molecular Biology, Book: “MERS Coronavirus”, ” DOI: 10.1007/978-1-0716-0211-9_9”

3. In Figure 4 the authors compare ct values. How did the authors guarantee that all PCRs had the same efficiency? Did they use a control RNA, or an in vitro transcript as standard?

In the PCR reactions, we used a control RNA and found that the Ct values for this control are within 5% difference between runs.

4. African dromedaries were sampled 2016 until 2018. The resident dromedary samples were exclusively collected in 2018. A direct comparison of e.g. RNA detection rates between African and Arabian dromedaries is therefore only possible for the year 2018. This limitation should be addressed.

The following statement was added in page 16-17, lines 343-347 “Although the sampling time frames of imported and domestic camels are different, as imported camels were sampled between 2016-2018 while domestic camels were only sampled in 2018, the results are still indicative of the prevalence of MERS-CoV in both groups and require further investigation and continuous screening programs to evaluate the seasonality of the infection.”.

---

## [Editor Report · Decision Letter 1]

22 Apr 2020

Cross-sectional prevalence study of MERS-CoV in local and imported dromedary camels in Saudi Arabia, 2016-2018

PONE-D-20-01538R1

Dear Dr. Azhar,

We are pleased to inform you that your manuscript has been judged scientifically suitable for publication and will be formally accepted for publication once it complies with all outstanding technical requirements.

With kind regards,

Stefan Pöhlmann, Ph.D.

Academic Editor

PLOS ONE
---

## [Editor Report · Acceptance letter]

14 May 2020

PONE-D-20-01538R1 

Cross-sectional prevalence study of MERS-CoV in local and imported dromedary camels in Saudi Arabia, 2016-2018 

Dear Dr. Azhar:

I am pleased to inform you that your manuscript has been deemed suitable for publication in PLOS ONE. Congratulations! Your manuscript is now with our production department. 

With kind regards,

on behalf of

Prof. Stefan Pöhlmann 

Academic Editor

PLOS ONE